# Dimensions of Tinnitus-Related Distress

**DOI:** 10.3390/brainsci12020275

**Published:** 2022-02-16

**Authors:** Petra Brueggemann, Wilhelm Mebus, Benjamin Boecking, Nyamaa Amarjargal, Uli Niemann, Myra Spiliopoulou, Christian Dobel, Matthias Rose, Birgit Mazurek

**Affiliations:** 1Tinnitus Center, Charité—Universitätsmedizin Berlin, Freie Universität Berlin and Humboldt-Universität zu Berlin, 10117 Berlin, Germany; petra.brueggemann@charite.de (P.B.); wilhelm.mebus@charite.de (W.M.); benjamin.boecking@charite.de (B.B.); nyamaa.amarjargal@charite.de (N.A.); 2Knowledge Management & Discovery Lab, Faculty of Computer Science, Otto von Guericke University, 39106 Magdeburg, Germany; uli.niemann@ovgu.de (U.N.); myra@ovgu.de (M.S.); 3Department of Otorhinolaryngology, Jena University Hospital, 07747 Jena, Germany; christian.dobel@med.uni-jena.de; 4Medical Department, Section of Psychosomatic Medicine, Freie Universität Berlin and Humboldt-Universität zu Berlin, Charité—Universitätsmedizin Berlin, 10117 Berlin, Germany; matthias.rose@charite.de

**Keywords:** tinnitus, comorbidity, psychological stress, factor analysis, predictors

## Abstract

Objectives: (1) To determine which psychosocial aspects predict tinnitus-related distress in a large self-reported dataset of patients with chronic tinnitus, and (2) to identify underlying constructs by means of factor analysis. Methods: A cohort of 1958 patients of the Charité Tinnitus Center, Berlin completed a large questionnaire battery that comprised sociodemographic data, tinnitus-related distress, general psychological stress experience, emotional symptoms, and somatic complaints. To identify a construct of “tinnitus-related distress”, significant predictive items were grouped using factor analysis. Results: For the prediction of tinnitus-related distress (linear regression model with R^2^ = 0.7), depressive fatigue symptoms (concentration, sleep, rumination, joy decreased), the experience of emotional strain, somatization tendencies (pain experience, doctor contacts), and age appeared to play a role. The factor analysis revealed five factors: “stress”, “pain experience”, “fatigue”, “autonomy”, and low “educational level”. Conclusions: Tinnitus-related distress is predicted by psychological and sociodemographic indices. Relevant factors seem to be depressive exhaustion with somatic expressions such as sleep and concentration problems, somatization, general psychological stress, and reduced activity, in addition to higher age.

## 1. Introduction

Chronic tinnitus denotes a subjective perception of sound in the absence of an external source. Whilst hearing loss can be associated with tinnitus, tinnitus-related distress arises mainly due to psychological factors [1,2,3].

There is no habituation to the harmless acoustic stimulus, but negative meaning is attributed to the percept and thereby becomes a disturbing permanent phenomenon—a distraction of attention no longer succeeds. The emerging stress of tinnitus perception and processing becomes tinnitus-related distress, the construct which will be further explored in this paper. Furthermore, distress must be considered and addressed in therapy; hearing–acoustic measures alone are not sufficient. The annual incidence rates (new cases) increased from 4.5 per 10,000 persons in 2002 to 6.6 per 10,000 in 2012 [4]. According to current European guidelines, up to 20% of the population suffer from chronic tinnitus for more than 3 months [5]. The assumptions regarding the complex pathophysiology of tinnitus have implications for treatment design. Tinnitus-related distress is multidimensional and amplified via concomitant auditory phenomena such as hearing loss or hyperacusis, psychological stress cycles, psychological comorbidities such as impaired concentration, sleep problems, negative thinking, or catastrophizing tendencies as well as depression or anxiety disorders [6,7,8,9,10]. Furthermore, tinnitus patients are often diagnosed with additional affective symptoms [3,9] and also cognitive changes [5,9,11,12,13,14]. Tinnitus is therefore not a purely acoustic problem. The accompanying restriction of the quality of life and comorbid symptoms in various areas are the result of multiple hearing-independent influencing factors, the grouping of which is the aim of the study.

The variety of existing hypotheses and models on the development, components, and maintenance of tinnitus-related distress corresponds to the heterogeneity of therapeutic approaches. In summary, however, a consensus emerges regarding multimodal treatment approaches that offer patients, in addition to education and counseling with attention to hearing loss and, if needed, hearing therapy measures, psychotherapy to stabilize and relativize tinnitus-related distress and associated comorbidities. In understanding tinnitus-related distress, it remains important to examine which of the manifold influences is most significant and must thus be given special consideration in diagnostics and therapy.

In this context, it is also clinically relevant to consider how to economically meas-ure multiple predictive factors for tinnitus-related distress for practitioners and pa-tients. Therefore, this paper analyses all questionnaires at item level (1) to identify items that significantly predict tinnitus-related distress and (2) to group these hear-ing-independent influencing factors to underlying dimensions which might be meas-ured more economically in future studies.

This study has been conducted on an extensive patient sample at the Charité Tinnitus Center, where the accrued patient group underwent a multimodal intensive tinnitus therapy over 7 days with elements of cognitive behavioral therapy, changing of acoustic attention, relaxation, and physiotherapy [14,15]. To account for the variety of accompanying symptoms and possible phenotypes of tinnitus-related distress, (1) sociodemographic data (including age, gender, education, marital status), (2) tinnitus-related distress (tinnitus questionnaire—TQ, recording of tinnitus quality—TLQ), (3) psychological distress and internal resources (PSQ—Perceived Stress Questionnaire; SWOP—recording of self-efficacy, optimism, pessimism), (4) depressive symptoms (ADS—General Depression Scale, BSF—Berlin Mood Questionnaire) and (5) somatic complaints (BI—Complaints Inventory; ISR—ICD10 Symptom Rating; SES—subjective perception of pain) were recorded.

In previous work of our group [16], predictors for tinnitus distress were determined using a different sample (including auditory data and cognitive tasks). In theory, the results should be similar in this sample; in this study, however, the regression will run exclusively on the individual items of the questionnaires in order to potentially build a questionnaire that integrates relevant items from various questionnaires. In contrast to previous publications on the same data set [17,18,19], tinnitus distress was predicted independently of therapeutic influences at the time of initial recording (t0). In the present study, the most important goal is the reduction of the items of different questionnaires to possible underlying content dimensions. In the future, this should enable better efficiency in diagnostics and specialization in therapy.

We formulated the following questions for this study:(1)Which sociodemographic and psychological items predict tinnitus-related distress at baseline?

For this purpose, all items were identified that significantly predicted tinnitus-related distress as measured by the Tinnitus Questionnaire (TQ-German version), and

(2)Which factors may underlie these items?

For this purpose, all significant items were summarized in a rotated factor analysis and subsequently interpreted.

Based on the understanding of underlying factors including the creation of an economic questionnaire instrument (items from research question (2) should allow the development of specific therapeutic strategies.

## 2. Materials and Methods

### 2.1. Sample

The study includes *N* = 1958 patients from the Tinnitus Center of the Charité—Universitätsmedizin Berlin from January 2011 to October 2015.

The entire study population consisted of participants in multimodal intensive tinnitus therapy for chronic tinnitus. The patients were already medically connected at the Tinnitus Center of the Charité—Universitätsmedizin Berlin prior to the 7-day multimodal intensive tinnitus therapy and continued to receive outpatient care there afterwards.

For the retrospective analysis of clinical data, patients with chronic tinnitus (existing for at least 3 months), an age of at least 18 years, and sufficient knowledge of German were included. Patients with severe psychiatric, central neurological, or addictive disorders were excluded. The clinical study was approved by the Ethics Committee of the Charité—Universitätsmedizin Berlin (EA4/137/20). All study participants voluntarily participated in the present data collection and were informed about the nature, significance, and aims of the scientific investigation. They had the opportunity to clarify open questions and gave informed consent for the collection, recording, processing, and publication of their data in anonymized form.

Descriptive analysis revealed a mean median age of 49 years and a balanced gender ratio. In total, 3/4 of the participants lived in partnership, about 70% were employed. Regarding education, 1% indicated that they were pupils or apprentices, 27% had completed an apprenticeship, 16% had a master’s or technical college degree, 3% were students, 50% had a college degree, and 3% indicated that they had no education. Regarding the duration of the tinnitus, 20% indicated less than half a year or half to 1 year each, 14% indicated 1 to 2 years, 16% indicated 2 to 5 years, and 30% indicated over 5 years. Over the course of their illness, 25% of patients consulted one physician, 31% consulted two physicians, 21% consulted three physicians, 11% consulted four physicians, 4% consulted five physicians, and 8% consulted over five physicians. (See also sociodemographic information in Table 1.)

### 2.2. Questionnaires

At baseline, data from the following eight questionnaires were obtained:(1)Tinnitus-related distress was assessed with the German version of the Tinnitus Questionnaire (TQ) [20]. The TQ [21] is an instrument for measuring tinnitus impairment or tinnitus-related distress (total score). The questionnaire contains 40 items with 3 response categories (2 = “agree”, 1 = “partially agree”, 0 = “disagree”) and the total score (TQ) is used as the dependent variable here;(2)The Perceived Stress Questionnaire (PSQ [22,23,24] contains 20 items to measure subjective stress experience within the last 4 weeks with the response scale 0 = “almost never”, 1 = “rarely”, 3 = “frequently” and 4 = “most of the time”. In the model we used 20 items;(3)The ICD-10 Symptom Rating (ISR [25,26] measures total psychological burden over the past 2 weeks via 29 items on a 5-point scale (0 = “does not apply”, 1 = “applies a little”, 2 = “applies quite a bit”, 3 = “applies to a great extent”, 4 = “applies extremely”);(4)The General Depression Scale (ADSL [27] contains 20 items measuring depressive symptoms in the last week, which were included in our model. The questionnaire is the German version of the globally used CES-D—Center for Epidemiological Studies Depression Scale [28]. Each item is rated on a 4-point ordinal scale. Clinically, we used the sum score, which indicates manifest depression from a score of 16;(5)The Self-Efficacy-Optimism-Pessimism Scale (SWOP [28]) contains 9 items that are rated on a scale from 1 = “agree exactly” to 4 = “disagree”. The 9 individual items were used in the calculation;(6)The Berlin Mood Questionnaire (BSF [29]) calculates a multidimensional mood model consisting of 30 items with 5 response options: 0 = “not at all”, 1 = “somewhat”, 2 = “quite”, 3 = “mostly”, and 4 = “total”;(7)The Berlin Complaint Inventory (BI [30]) contains 57 items from the fields of fatigue, upper abdominal symptoms, limb pain, heart problems, and a sum score for total complaint pressure. Responses are on a 5-point ordinal scale with 0 = “not at all”, 1 = “hardly”, 2 = “to some extent”, 3 = “considerably” to 4 = “strongly”. Responses to the 57 individual items were included in the calculation;(8)The Pain Perception Scale (SES [31]) results in sum scores for affective and sensory pain from 24 items with the 4 response options from 1 = “does not apply” to 4 = “applies exactly”. All 24 items were included in our analysis.

Most questionnaires were answered on a Likert scale. The ordinal values were treated as continuous values in the analysis. Categorical values, e.g., gender, marital status, or educational level, were transformed into dichotomous variables. A brief overview of all included variables and the descriptors of the sample are presented in Table 1.

### 2.3. Statistics

Statistical analysis were performed using IBM SPSS Statistics 26 for Microsoft Windows. Statistical analyses were based on a significance level of *p* < 0.05, or *p* < 0.01. To answer research question 1, the prediction coefficients of all items from the questionnaires used for stress (PSQ, SWOP), depressive symptoms (ADS, BSF), and somatic complaints (BBI, ISR, and SES) and the influence of the dichotomized demographic social data variables were calculated using an automatic linear regression model. The TQ total score at baseline was specified as the dependent variable.

The significant items in the regression model were then included in an exploratory factor analysis to answer research question 2 in order to identify underlying factors of tinnitus-related distress.

The main component analysis was chosen as the extraction method. To improve the interpretability of the factors, a Varimax rotation with Kaiser normalization was chosen. It was determined which items loaded on which factors, and item loadings of >0.5 were considered significant.

## 3. Results

### 3.1. Predictors of Tinnitus-Related Distress/Regression Model

A total of 187 items from 7 questionnaires and 10 sociodemographic questions were included in the regression analysis. All included questionnaire items explained about 70% of TQ variance.

A total of 36 items significantly predicted the TQ total score (8 from PSQ, 2 from SWOP, 4 from ADS, 2 from BSF, 6 from ISR, 3 from BI, and 6 from SES, and 5 sociodemographic items), see Table 2. The importance of each predictor was rather equal and low (values between 0.09 and 0.01). The top 10 (with coefficients greater than 0.02) are shown in Figure 1.

Age has the highest predictive value, followed by the number of physicians contacted (somatization tendency). For the predictive social data, the following result was obtained: tinnitus-related distress is higher with increasing age. The more physicians contacted for tinnitus before starting therapy, the higher the experienced distress from tinnitus. Tinnitus-related distress decreases when psychotherapy is used. Concentration problems (from ADS), relaxation problems (PSQ), sleep disturbances (ISR), and pain perception (SES) are also statistically significant predictors; furthermore, from the PSQ, reduced feeling of joy. From the Berlin Mood Questionnaire, increased disturbance is significant. From the Complaints Inventory (BI), restricted sleep.

Tinnitus-related distress is higher with higher levels of depression, tension/fatigue, increased affective pain experience, and various somatizations (including sleep disturbances). Employment and educational level have a positive effect on tinnitus-related distress. Contact to several physicians before the current treatment indicates more complex, possibly more chronified symptoms, and tinnitus-related distress is higher. The number of physicians consulted can be taken as an indication of somatization. An initiated psychotherapy significantly reduces experienced tinnitus-related distress.

### 3.2. Dimensional Reduction for Tinnitus-Related Distress via Factorization

First, a factor model for item reduction without factor limitation was calculated, resulting in 7 factors. The number of factors to be extracted was initially not limited, and 7 factors could be extracted following Scree-Plot inspection and review of the Kaiser–Guttmann Criterion (equivalent value >1); however, only the first two showed significant variance shares. In order to improve the interpretability of the factor model, it was further decided to focus on factors which accounted for more than 5% of the total variance in the model and which together accounted for more than 50% of the total variance. This resulted in a 5-factor model (Figure 2).

The results of the factor analysis are summarized in Table 3.

The component matrix after rotation results in the following eigenvalues (Table 4). For greater readability, only eigenvalues greater than 0.3 are considered in the representation and sorted by size.

Factor one (“stress”), with the highest variance explanation, combines items on stress perception from the PSQ, depressive experience from ADS with mood states (ISR, BSF), and pessimism (SWOP). The second factor (“pain experience”) explains 12% of variance after rotation. It mainly consists of items from the SES concerning the subjective experience of pain. The third factor (“fatigue”) still explains 11% of variance and includes mainly distress-associated somatic symptoms (sleep, concentration problems, dizziness). The fourth factor (“autonomy”) explains almost 5% of variance and refers to external stressors with less control experience and possible feelings of dependence (employees, little time for oneself, fear of not being able to reach one’s goals) and concerns rather younger patients. The 5th factor (“education level”) with 4.5% variance explanation refers mainly to social variables and concerns older patients with higher education and fewer physician contacts before the start of treatment.

In summary, a total of 36 items, 31 from the questionnaires and 5 sociodemographic items predicted 70% of variance in tinnitus-related distress at baseline (Table 2). An explorative factor analysis revealed five underlying factors that were labelled “stress”, “pain experience”, “fatigue”, “autonomy” and “education level”. The first three factors explain the highest amount of variance and show high conceptual similarity to depressive experiences (see also Table 2). A short conclusion of the analysis and results is shown in Figure 3.

## 4. Discussion

The regression analysis of the items with tinnitus-related distress (TQ total score) as the dependent variable shows the prediction of tinnitus-related distress via psychological distress/difficulties to relax, affective and body symptoms, and social data such as age. Especially, items pertaining to psychological stress and depressive symptoms (ISR, ADS) increased experience of tension (PSQ), but also affective experience of pain (SES) as well as physical stress (BBI), mood (BSF), and self-efficacy (SWOP) seem to be relevant for experienced tinnitus-related distress. Social factors such as age and education level are also shown to be relevant in this study. In a previous study by the working group [16], stress (tension) was described as the most important predictor of tinnitus distress. In this study, including cognitive tasks and hearing data, decreased hearing and concentration were also described as predictive of tinnitus distress, as well as social data (e.g., number of doctors visited) and coping strategies (pessimism).

In a study of risk factors for the chronification of tinnitus, Holgers and colleagues in a review [32] found depression and physical immobility to be predictors of maintenance of tinnitus-related distress; hearing loss was a moderate predictor. Based on these data, they developed a model that explained tinnitus-related distress from a mixture of depressive and anxious symptoms, along with physical problems and hearing restriction. In our synopsis of the data from the study group, we follow this model, whereby tinnitus-related distress also leads to cognitive impairment independent of concomitant hearing impairment [13]. Whether these cognitive impairments are part of a depressive event remains to be seen. In a previous study of the working group, Niemann [19] used machine learning to calculate items that are essential for predicting the success of the multimodal tinnitus therapy described here.

The results of the factor analysis show five fundamental dimensions of tinnitus distress. In our opinion, this is the first empirical demonstration (using a large dataset including a wide range of psychological questionnaires and patient data) that tinnitus-related distress is multimodal. In addition to a confirmation of relevant models in this direction, this is of great importance for adequate therapy planning.

The aim of this work was to uncover constructs that underlie tinnitus-related distress that can be used for diagnostics and therapy. The results of this dataset yielded 5 factors which we labelled “stress”, “pain experience”, “fatigue”, “autonomy”, and “education level”. We consider these 5 factors as underlying constructs of tinnitus-related distress.

The significant role of stress in the maintenance and chronification of tinnitus has been described several times by our group [12,33]. Factors 2 and 3 are not absolutely distinct from factor 1, and we see all three factors co-occurring with depressive symptoms. For example, factor 2 (pain) may well be seen as a consequence of exhausted depressive or anxious symptomatology [34]. The level of education (factor 5) or even (not) experienced autonomy (factor 4) may, in turn, have an impact on the use of coping mechanisms and the elaboration of individual disease models. The factors stress, autonomy, and level of education should be taken into account in therapy through specific stress reduction programs. Within the framework of cognitive behavioral therapy, if possible in a group, the causes of stress (including basic personal beliefs and existing coping strategies) should be addressed and changed. Stress is reduced via a changed evaluation. Pressure to perform and perfection become visible as a result of individual beliefs and patients learn to distinguish between internal and external stressors. Too much striving for autonomy (not being able to accept help) must be reduced just as much as not experiencing enough self-control. This can be done in a group, but also in individual discussions or accompanying psychotherapy. The level of education seems to play a role not only in performance pressure but also in coping strategies. A tinnitus therapy program should therefore definitely evaluate existing strategies for dealing with stress and, if necessary, encourage the development of alternative coping strategies.

The factors of pain and exhaustion should also be discussed as part of the CBT group discussions or accompanying therapies. Changes can be initiated here via revaluation or activation. However, it is also essential to include body-related procedures in tinnitus therapy. This means relaxation techniques, mindfulness-related procedures, and/or physiotherapy. In the case of severe pain or depressive illness, tinnitus sufferers must also be advised by a doctor with regard to pharmacological support. All this should be components of a multimodal tinnitus therapy. Additionally, as early as the diagnostic phase, comprehensive counseling by an ENT doctor should, for example, point out these possibilities and make individual recommendations.

The extent to which the 5 factors are truly independent of each other in terms of content must remain an open question at this time. Future multimodal therapies may have to offer therapy strategies in a building-block principle that positively influence these dimensions. These will then be used in a way that is individually tailored to the patient. Cognitive behavioral therapy continues to play a critical role. This can contribute not only to altered perception and reappraisal in tinnitus, but also to reduction of depression and anxiety, alteration of pain perception, and alleviation of somatic distress or harnessing of resources. Relaxation skills to reduce somatic symptoms of distress and feelings of exhaustion, such as sleep problems, are taught via relaxation techniques, mindfulness-based methods, or even body-based methods in the context of physical therapy and sports. Tinnitus patients should at least be made aware of this component. Overall, therapy is about building up resources and protecting against excessive demands (distress).

The identified items in the regression model allow a better diagnostic approach to the expression of multimodal tinnitus-related distress. The significant individual items can be incorporated into a questionnaire that effectively and economically captures the important facets of tinnitus-related distress. Further work in this regard is planned.

Limitations: The predictors presented on tinnitus-related distress are not valid independent of the questionnaires used. It is known that hearing ability and cognitive strategies in terms of top-down regulation also lead to altered tinnitus perception, which was not investigated here. Due to the accruing clinical sample, no control group is available.

All included items were based on reliable and valid self-report instruments, however, international comparability has yet to be demonstrated. For the regression, we made the statistical assumption of a continuous distribution of the ordinal items. Moreover, also to be factored in is that high intercorrelation of self-assessment tests always result. The tinnitus-related distress to be predicted (measured by the TQ total score) represents a psychological measurement construct, whereas overlaps to the measurements of the other questionnaires certainly exist. To what extent the identified 5 factors are truly clinically independent of each other remains to be shown by future research.

## 5. Conclusions

Overall, in this large sample with tinnitus patients, underlying factors for tinnitus-related distress mainly reflect stress-correlated aspects of an exhausted–depressive experience with somatization, pain perception, experienced control alongside demographic aspects such as age and education level (significant for coping strategies). These factors must be measured and therapeutically considered in multimodal therapies for chronic tinnitus.

## Figures and Tables

**Figure 1 brainsci-12-00275-f001:**
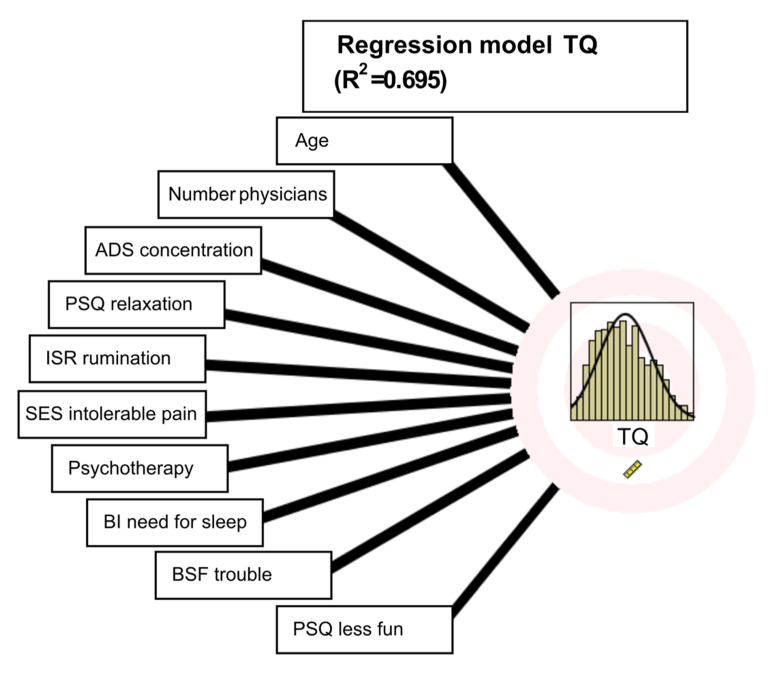
Regression model results for the top 10 predictors.

**Figure 2 brainsci-12-00275-f002:**
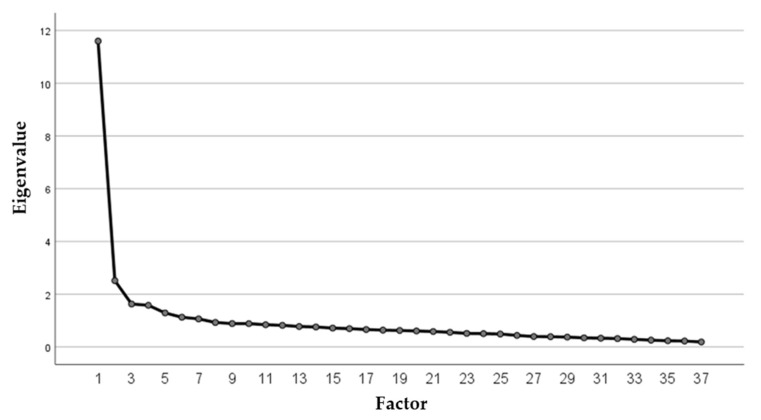
Factor plot (equivalent value and number of factors)—definition of 5 factors.

**Figure 3 brainsci-12-00275-f003:**
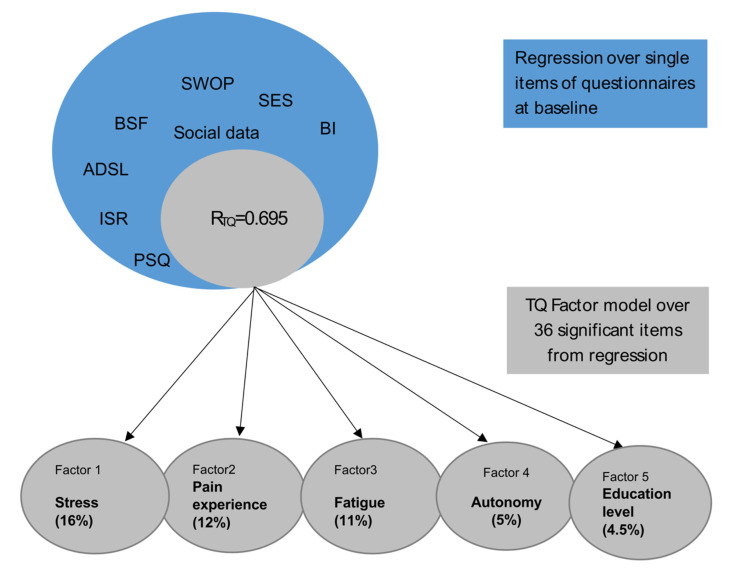
Conclusion of analysis model and resulting factors.

**Table 1 brainsci-12-00275-t001:** Sample characteristics and questionnaires (*N* = 1958).

Variables/Sociodemographic Information	Mean (st.dev.)/Percent
Sex	women: 50%
Age	49.4 (11.8)
Partnership	yes: 72%
Graduation	low: 11% middle: 39% high: 49%
Employment Duration of tinnitus Number of physicians consulted	yes: 74.7% 2.7 years (1.9) 2.95 (1.6)
**Variables**	**Mean (st.dev.)**
Total TQ: tinnitus-related distress Emotional distress Cognitive distress Psychological distress Intrusiveness Auditory perceptual difficulties Sleep disturbances Somatic complaints	35.2 (17.2) 9.5 (5.6) 5.9 (4.1) 15.4 (9.3) 9.6 (3.8) 4.8 (3.6) 3.3 (2.5) 2.1 (1.9)
PSQ-20 total: perceived stress Worries Tension Joy Demand	44.8 (18.3) 27.7 (22.7) 55.9 (22.6) 49.5 (23.2) 48.8 (22.2)
SWOP Self-efficacy Optimism Pessimism	28.0 (5.8) 27.7 (7.6) 21.2 (7.2)
ADSL total depression	15.6 (11.3)
BSF berlin mood questionnaire Elevated mood Positive mindset Anger Anxious depression Fatigue Apathy BI Berlin complaint inventory Overall complaints Fatigue Abdominal symptoms Limb pain Heart symptoms	1.6 (1.0) 2.2 (0.77) 0.7 (0.71) 1.08 (0.92) 1.49 (1.01) 0.56 (0.67) 22.69 (15.32) 8.53 (5.89) 2.76 (3.39) 8.40 (5.39) 3.00 (3.68)
SES pain perception scale Affective pain Sensoric pain	23.34 (9.86) 13.57 (4.95)
ISR ICD-10 Symptom Rating: total score Depression Anxiety Obsessive–compulsive syndrome Somatoform syndrome Eating disorder syndrome Additional items score	0.78 (0.58) 1.10 (0.92) 0.91 (0.90) 0.79 (0.85) 0.59 (0.78) 0.66 (0.81) 0.71 (0.55)

TQ = Tinnitus questionnaire, PSQ = Perceived Stress Questionnaire, SWOP = Self-Efficacy-Optimism-Pessimism Scale questionnaire, Pain Perception Scale, ADS = General Depression Scale, BSF = Berlin Mood Questionnaire, BI = Berlin Complaint Inventory, SES = Pain Perception Scale, ISR = ICD-10-Symptom-Rating.

**Table 2 brainsci-12-00275-t002:** Significant predictors from regression model for tinnitus-related distress.

Sociodemographic Items that Significantly Predict TQ		
Age		
Number of physicians consulted		
Psychotherapy		
Graduation		
Employment		
**Psychological Questionnaire Items that Significantly Predict TQ**		
**Questionnaire**	**Item Number**	**Item Text**
PSQ	03	You feel that too many demands are being made on you.
PSQ	06	You fear that you will not be able to achieve your goals.
PSQ	20	You feel discouraged.
PSQ	21	You are having fun (-).
PSQ	22	You are afraid of the future
PSQ	26	You feel mentally exhausted.
PSQ	27	You have trouble relaxing.
PSQ	29	You have enough time for yourself (-).
SWOP	03	I have no trouble achieving my intentions and goals.
SWOP	06	Things never turn out the way I want them to.
ADS	05	During the last week, I had trouble concentrating.
ADS	06	During the last week, I have been depressed.
ADS	11	During the last week, I slept poorly.
ADS	14	During the last week, I felt lonely.
BSF	12	I feel troubled.
BSF	29	I feel exhausted.
ISR	02	I no longer really enjoy things that I normally like to do.
ISR	10	I resist repetitive, nonsensical thoughts or actions, even if I don’t always succeed.
ISR	18	I have difficulty concentrating.
ISR	20	I have problems sleeping.
ISR	24	I have psychological problems due to severe everyday stresses (e.g., serious illness, loss of job, or separation from partner).
ISR	27	I feel impaired in the exercise of my sexuality.
BI	07	Excessive need for sleep
BI	10	Dizziness
BI	27	Headache
SES	01	I experience my pain as tortourus.
SES	02	I experience my pain as horrible.
SES	07	I experience my pain as dreadful.
SES	09	I experience my pain as heavy.
SES	13	I experience my pain as intolerable.
SES	17	I experience my pain as burning.

Notes: PSQ = Perceived Stress Questionnaire. SWOP = Self-Efficacy-Optimism-Pessimism Questionnaire. ADS = General Depression Scale. BSF = Berlin Mood Questionnaire. BI = Berlin Complaint Inventory. ISR = ICD-10 Symptom Rating. SES = Pain Perception Scale.

**Table 3 brainsci-12-00275-t003:** Total variance explained over 5 factors.

Component	Initial Eigenvalues	Total% of Variance	Cumulated%
1	11.597	31.342	31.342
2	2.514	6.794	38.136
3	1.626	4.395	42.536
4	1.578	4.264	46.795
5	1.287	3.479	50.275
Extraction method: principal component analysis			
**Sums of Squared Factor Loadings for Extraction**			
**Total**	**% of variance**		**Cumulated%**
11,597	31.342		31.342
2514	6.794		38.136
1626	4.395		42.531
1578	4.264		46.795
1287	3.479		50.274
**Rotated sum of squared loadings.**			
**Total**	**% of variance**		**Cumulated%**
6040	16.324		16.324
4565	12.338		28.662
4130	11.163		39.825
1806	4.882		4.,707
1646	4.449		49.156

**Table 4 brainsci-12-00275-t004:** Eigenvalues of the items on the 5 factors after rotation.

Rotated Component Matrices					
Component	1	2	3	4	5
PSQ 20	0.745				
PSQ 22	0.686				
PSQ 21	−0.679				
ISR 02	0.634		0.361		
ISR 24	0.632				
ADSL 06	0.622		0.417		
ADSL 14	0.609				
BSF 12	0.595		0.390		
PSQ 06	0.574			0.332	
PSQ 26	0.559		0.463		
ISR 27	0.557				
SWOP 06	0.513				
SWOP 03	−0.481				
PSQ 03	0.455				
social item 24 (psychotherapy)	0.429				
ISR 10	0.323				
SES 02		0.831			
SES 13		0.806			
SES 09		0.799			
SES 07		0.789			
SES 01		0.764			
SES 17		0.584			
BI 27		0.437	0.435		
ADSL 11			0.796		
ISR 20			0.780		
BSF 29	0.457		0.626		
BI 07			0.573		
ADSL 05	0.401		0.571		
PSQ 27	0.392		0.508		
ISR 18	0.464		0.505		
BI 10		0.322	0.372		
social item 18 (employment)				0.648	
PSQ 29				0.620	
social item 10 (profession, foreman)					0.707
age				−0.505	0.515
social item 09 (graduation)					0.483
social item 25 (number of physicians consulted)					−0.379

Extraction method: Principal component analysis. Rotation method: varimax with Kaiser normalization. The rotation converged in 6 iterations.

## Data Availability

The data presented in this study are available on request from the corresponding author. The data are not publicly available due to institutional rules.

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
