# Peer review of "Dimensions of Tinnitus-Related Distress"

_brainsci, 2022, doi:10.3390/brainsci12020275_

Round 1

Reviewer 1 Report

This article explores predictive psychosocial contributors of tinnitus-related distress measured by TQ in a large patient sample. A previous study by some of the authors addressed a similar set of research questions by examining the relationship between TQ with a similar set of questionnaire scores (predictors) in 531 tinnitus patients, using correlation and multiple regression (eg reference 16). Previous published findings showed that depression, stress scores and hearing loss had particularly high predictor values. The current study reported similar results with depressive exhaustion and somatic expressions being relevant to tinnitus-related distress.

Major comments

1) Given that the current study reports an incremental finding based on part of the same dataset, the authors should i) give more details in the Introduction about this underpinning work. Ii) be clear what research gap is addressed in the present study, and iii) clarify what fresh insights are gained from the results. For example, Research question 1 ‘Which items predict tinnitus-related distress at baseline?’ seems to be a repeat of the previous work conducted by the authors.

2) One could argue that the premise of the study is somewhat circular. The TQ score which is used as a measure of tinnitus-related distress is a composite of questions about emotional distress, cognitive distress, psychological distress, intrusiveness, auditory perceptual difficulties, sleep disturbances, and somatic complaints. It is therefore unsurprising that psychological distress, sleep disturbances and somatic complaints as measured by other questionnaires are correlated with the global TQ score and hence significant in the regression analysis. These findings do not provide such novel insight because the analysis is somewhat circular.

3) Page 2 line 142 “The ordinal values were treated as continuous values in the analysis”. Ordinal variables with five or more categories are often used as continuous without any harm to the analysis (Johnson & Creech, 1983; Norman, 2010; Sullivan & Artino, 2013; Zumbo & Zimmerman, 1993), but the PSQ, ADSL, SWOP, and SES are described with only 4 categories. Can the authors provide greater statistical justification for treating this ordinal scale as continuous? What are the potential risks for the data interpretation if this assumption is not correct?

4) Section 2.3 ‘Statistics’ There are a number of different regression models Can the authors give more details of the regression method used?

5) Regression often becomes less effective the larger the number of potential explanatory variables; some real explanatory variables may happen to not be statistically significant, while nuisance variables may be coincidentally significant. As a result, the model may fit the data well in-sample, but do poorly out-of-sample.

See https://journalofbigdata.springeropen.com/articles/10.1186/s40537-018-0143-6

This point is relevant because a lot of the challenges with this type of data driven work is that different studies report different findings. What would be more interesting is if the authors replicate some of their earlier findings in a new dataset. I would strongly invite the authors to consider the suggestion to repeat some of the multiple regression without those same patients who had contributed to one of the earlier studies (eg references 16-19). If the same predictive effects are obtained, that would be much more compelling to the scientific case.

6) The premise of the regression model is that the ‘predictor variables’ determine tinnitus symptom severity. In other words, it assumes a directional causal relationship. While it’s reasonable to interpret the variable ‘age’ in this way, I would disagree that the same is true for ‘number of physicians contacted’. Surely tinnitus severity drives the person to consult a medical specialist, and not the other way round?

8) The significant items in the regression model were then included into an exploratory factor analysis to identify underlying factors of tinnitus-related distress. Again, I have a concern that this is a circular analysis because it is based on the same dataset of 1958 patients.

Minor comments

Line 35 : generell

Line 168: Variance if all

Line 226: fatigue

Line 254: These findings represent a confirmation and extension of the predictors of tinnitus-related distress already described by the working group in 2016 (16).

Reviewer 2 Report

Dear Author, I read with interest your manuscript. It is well done and useful for the scientific literature for the high number of patients examined. Only one thing that you can improve in the discussion. It is not so clear who your research can improve the treatment of the tinnitus.  Can you explain in detail which of the 5 factors, stress, pain, fatigue, autonomy, education level has to be treated? Can you improve this aspect in discussion?

Round 2

Reviewer 1 Report

I thank the authors for considering my comments and for making changes to the manuscript. It seems that most of the changes made are in the Introduction and Discussion, with minimal change to the Methods and Results. In my original review, I raised a number of substantive statistical questions which I feel were either rebutted or remain still not adequately address by the authors. In fact, the authors now explain that they used the automatic regression option in SPSS which actually takes away  experimenter control over decisions about the statistical analysis.

First, multiple linear regression analysis makes two key assumptions: i) that the residuals are normally distributed, and ii) that the independent variables are not highly correlated with each other. The authors do not confirm that these assumptions were checked and are met. With a total of 187 items from 7 questionnaires and 10 social data questions included in the regression, I would find it surprising if none of the items were correlated and that some show multicollinearity. For example, the authors have added the comment that one of the predictors (“the number of physicians consulted“) can be taken as an indication of somatization. Yet there are already a number of somatization items from the regression model which are claimed to also be predictors. I might expect this redundancy to be indicative of multicollinearity. If this is the case, then the data need to be pre-treated appropriately before modelling.

Second, the automatic linear regression model in SPSS is a way to enable the statistical software to specify its best guess at the model the researcher wants using role information, and uses a standard forward regression model. I have two major concerns. Decision making is taken out of the hands of the researcher and so decisions are not always well informed and evidence based. Some disadvantages of the automatic method are that the researcher hands over to SPSS control over model specification, and over how to deal with missing data and outliers. In addition, the default approach is a forward stepwise regression method unless “all predictors“ are selected. Forward selection has significant drawbacks for large datasets, including the fact that each addition of a new predictor may render one or more of the already included predictors non-significant (p-value>0.05). Finally, the outputs are limited to graphical descriptions (eg Figure 1), which are not appropriate for a reputable scientific journal.

Why can the authors not repeat the regression modelling in SPSS using manual researcher guided selection of the model specification, so that the methods and model specification can be fully described, and results fully reported? This would provide greater transparency and hence promote replication by others.

Finally, some of the additional text in the Introduction does not make sense: 1. “On the one hand, the results should be similar in this sample. In this study, however, the regression will run exclusively on the individual items of the questionnaires. This in order to develop perspective a questionnaire that integrates items relevant to tinnitus distress from various questionnaires“. 2. “Perspective, from the understanding of underlying factors, the creation of specific therapeutic strategies and an economic questionnaire instrument (items from research-question 2) should become possible.“
